# Fumonisins in African Countries

**DOI:** 10.3390/toxins14060419

**Published:** 2022-06-19

**Authors:** Tapani Yli-Mattila, Leif Sundheim

**Affiliations:** 1Molecular Plant Biology, Department of Life Technologies, University of Turku, FI-20014 Turku, Finland; 2Norwegian Institute for Bioeconomy Research, P.O. Box 115, N-1431 Ås, Norway; leif@sundheim.no

**Keywords:** fumonisins, Africa, *Fusarium verticillioides*, *F. proliferatum*

## Abstract

Maize and other cereals are the commodities most contaminated with fumonisins. The maize acreage is increasing in Africa, and the maize harvest provides important foods for humans and feeds for domestic animals throughout the continent. In North Africa, high levels of fumonisins have been reported from Algeria and Morocco, while low levels have been detected in the rather few fumonisin analyses reported from Tunisia and Egypt. The West African countries Burkina Faso, Cameroon, Ghana, and Nigeria all report high levels of fumonisin contamination of maize, while the few maize samples analysed in Togo contain low levels. In Eastern Africa, high levels of fumonisin contamination have been reported from the Democratic Republic of Congo, Ethiopia, Kenya, Tanzania, and Uganda. The samples analysed from Rwanda contained low levels of fumonisins. Analysis of maize from the Southern African countries Malawi, Namibia, South Africa, Zambia, and Zimbabwe revealed high fumonisin levels, while low levels of fumonisins were detected in the few analyses of maize from Botswana and Mozambique.

## 1. Introduction

Fumonisins are mycotoxins causing serious contamination of foods and feeds [1,2]. The chemical structure of the fumonisin B_1_ was determined independently in South Africa [3,4] and New Caledonia in 1989 [5]. Fumonisin B_1_ is a diester of propane-1,2,3,-tricarboxylic acid and 2-amino-12,16-dimethyl-3,5,10,14,15-pentahydroxyicosane [3,6]. The B series of fumonisins B_1_, B_2_, and B_3_ account for most of the fumonisins that contaminate grain samples. Fumonisin B_4_ is a minor metabolite. Fumonisins C_1_, C_2_, and C_3_ are the main fumonisins produced by *F. oxysporum*. In addition, there are P- and A-series of fumonisins, which are minor metabolites [7]. The regulatory limits for fumonisins in maize and its by-products established by the European Union and Food and Drug Authority to prevent exposure of individuals to these fungal toxins is 200–4000 μg kg^−1^ [8]. The fumonisins content in unprocessed maize for human consumption is not permitted to exceed 4000 μg kg^−1^ in the EU [9].

Leukoencephalomalacia was first described in horses and other farm animals in USA more than one hundred years ago [10]. A connection was found between the disease and mouldy maize grain, and it was possible to cause the disease in horses and pigs by feeding them mouldy maize from disease outbreak areas. Later outbreaks of the disease were reported in South America, China, Africa, and Europe [10]. Fumonisins have also been connected to pulmonary oedema syndrome of pigs in USA [11], oesophageal cancer in South Africa [12] and neural tube effects in USA and Mexico [10,13]. The disease was first connected to *Fusarium verticillioides* in USA and subsequently in South Africa [10]. The aim of the present paper was to collect the data for the fumonisin situation in different areas of Africa and to discuss, how to control fumonisin intake.

## 2. Results

### 2.1. Fumonisin Producing Fungi

Fumonisins are mainly produced by species of *Fusarium*, while closely related compounds are produced by species of *Altenaria* [14]. High levels of fumonisins production have been found mainly in *F. verticillioides* and *F. proliferatum* strains of the *Gibberella fujikuroi* species complex. Fumonisin biosynthetic (*FUM*) gene clusters have been reported in *F. verticillioides*, *F. proliferatum*, *F. globosum, F. nygamai* and in a single strain of *F. oxysporum* [15,16]. It has also been reported that strains of *Aspergillus niger* are able to produce FB_2_ [17,18]. Species-specific polymerase chain reaction (PCR) is commonly used to identify species inside fungus complexes such as *F. verticillioides* and *F. proliferatum* [19], while quantitative PCR (qPCR) can be used for quantification of these species in cereal grains and kernels to estimate the risk for high fumonisin levels [16]. *Fusarium verticillioides*, the main fumonisin-producer, is a filamentous ascomycete that typically causes maize seedling blight and root, stalk, and ear rots, but the pathogen is also commonly associated with maize as an endophyte without any development of symptoms [20,21,22]. Strains of *F. verticillioides* from banana do not have the FUM gene cluster, and they do not produce fumonisins. These banana strains exhibit reduced virulence on maize seedlings [23]. In a region of South Africa with frequent occurrence of human oesophageal cancer, the disease was related to high levels of maize consumption [24].

The most common and important sources of fumonisin contamination in humans and animals are cereals (rice, wheat, barley, maize, rye, oat, and millet). Maize and maize-based products are the foods most infected by FB_1_ [10]. In the Transkei region of South Africa, the association between consumption of fumonisin contaminated maize and incidence of human oesophageal cancer was confirmed [25]. Purified fumonisins fed to a horse caused equine leukoencephalomalacia [26]. The most toxic of the fumonisins FB_1_, was recently evaluated, and the conclusion reached that genotoxicity and epigenetic properties have not yet been clearly elucidated [27]. The International Agency for Research on Cancer classified fumonisin FB_1_ as a group 2B carcinogen, probably carcinogenic in humans [28].

From infected host plants and commodities, several fumonisin analogues were described that grouped into the A, B, C and P series. In naturally infected maize only fumonisins of the B series are of significance. The types FB_1_, FB_2_ and FB_3_ are the most abundant of the naturally occurring fumonisins. *Fusarium verticillioides* and *F. proliferatum* are the most prevalent fumonisin producing pathogens in maize, which has become a major staple food for rural communities in Southern Africa [26]. Mycotoxin contaminated maize and other commodities have negative effects on human health and on the health of domestic animals. Additionally, mycotoxin contamination reduces the competitiveness of commodities on the export market. Intervention strategies should provide support for capacity building and increase the public awareness of the problem [28].

Host plants are frequently infected with both fumonisin and zearalenone. Cereals and cereal-based commodities from Botswana, Kenya, Malawi, Mozambique, South Africa, Tanzania, Uganda, and Zimbabwe were analysed for total fumonisin. In 37 of 40 samples, FB_1_ was detected in concentrations ranging from 20 to 1910 µg kg^−1^, and the total content of fumonisins (FB_1_, FB_2_ and FB_3_) ranged from 20 to 2735 µg kg^−1^. Samples from Zimbabwe had the highest total fumonisins content with 2735 µg kg^−1^. Only 5 of the 40 samples contained zearalenone in concentrations from 40 to 400 µg kg^−1^. There was no correlation between the levels of fumonisin and zearalenone in the samples [29]. Maize production in 10 African countries is presented in Table 1.

### 2.2. Fumonisins in Northern African Countries

A survey in Algeria showed that fumonisins frequently contaminate maize in the country. Analyses of cereal lots revealed that the toxins were present in rather high concentrations, which represents a health risk for the consumers. Fumonisin FB_1_ was present in 29 out of 30 samples, and FB_2_ was detected in 27 of 30 samples. The mean concentration of fumonisins in positive samples were 14,812 µg kg^−1^ for FB_1_ and 8603 µg kg^−1^ for FB_2_. Fumonisins were not detected in barley, rice, and wheat samples [31]. Birds are sensitive to fumonisin contamination. When samples from bird feed were analysed, 11.6% had a fumonisin content below 400 µg kg^−1^, 53.6% had a fumonisin content below 3000 µg kg^−1^, and 34.78% had a fumonisin content above 3000 µg kg^−1^. Fumonisin contamination of maize represents a threat to humans, domestic animals, and birds in Algeria [32]. Analysis from a survey of 120 cereal samples collected in the markets of Algeria revealed rather high concentrations of fumonisins. The range of concentrations of FB_1_ + FB_2_ was 289 to 48,878 µg kg^−1^, which represents a health risk for consumers and domestic animals. The results indicate the need for continuous control of fumonisins in the maize harvest in Algeria [33].

In Egypt, 25 samples of species of *Fusarium* were isolated from maize and wheat, and 21 of those were identified as *F. verticillioides*. Eight of the maize isolates produced fumonisins, while none of the five isolates obtained from wheat did [33]. Another study from 20 districts in Egypt revealed that the fumonisin content in maize was only 33 µg kg^−1^, while that of rice was 1014 µg kg^−1^ [34]. When samples of date palm fruits were analysed in Egypt, 7% of the samples contained FB_2_ in the range of 4.99 to 16.2 µg kg^−1^ [35]. To determine the mycotoxin status of the Egyptian crops, 57 samples from maize and maize-based products from several districts in the country, were analysed for fumonisins. Fumonisin FB_1_ was detected in 80% of the yellow maize samples, 33.3% of the corn flour samples, and in 28.57% of the popcorn samples. The concentration of FB_1_ varied from 10 to 780 µg kg^−1^. Baking balady bread from the contaminated corn meal reduced the FB_1_ content by 27.6% [36].

Twenty samples each of maize, wheat and barley were collected on the markets of Rabat and Salé in Morocco and analysed for fumonisins. The average fumonisin concentration was 1930 µg kg^−1^ and the co-occurrence of several mycotoxins was determined [37]. Results from the analysis for fumonisins FB_1_, FB_2_ and FB_3_ in 48 breakfast cereals and 20 infant cereals from Morocco revealed their detection in 18 breakfast and 2 infant cereals. The most contaminated commodities were maize-based cornflakes and breakfast cereals produced from rice, maize, and cacao. A fumonisin content of 228 µg kg^−1^ was detected in samples from a breakfast cereal [38]. Analysis of products on the market in Morocco revealed that in bread the levels of fumonisins ranged from not detected (ND) to 133.77 µg kg^−1^, in wheat from ND to 484.78 µg kg^−1^, in biscuits from 25.43 to 188.71 µg kg^−1^, and in two samples of semolina, fumonisins content were ND and 42.79 µg kg^−1^ [39]. In field trials, conventional tillage compared to no-till treatments did not affect the fumonisin contamination, but increased nitrogen fertilization increased the fumonisin levels. In Morocco, the average fumonisin content of harvested maize were 516 µg kg^−1^ in 2000, 5846 µg kg^−1^ in 2001 and 3269 µg kg^−1^ in 2002 [40].

In Tunisia, fumonisins were detected in 10.5% of 180 maize food samples with levels from 70 to 2130 µg kg^−1^. All contaminated samples contained FB_1_ and 31.5% of the samples contained FB_2_. Analysis of 15 maize feed samples, revealed that 86.6% were contaminated with fumonisins in concentrations ranging from 50 to 2800 µg kg^−1^. A significant correlation was detected between level of *F. verticillioides* and fumonisin contamination in maize [41]. Another study in Tunisia revealed that fumonisins were found in 20.83% of the wheat samples, 40% of the barley samples and in 57.14% of the maize samples. There was a strong correlation between the prevalence of *F. verticillioides* and the fumonisin contamination [42].

### 2.3. Fumonisins in Western African Countries

Rather high fumonisin contamination of maize has been detected in Benin. Local fumonisin levels were reported to range 8240–16,690 µg kg^−1^, with variation from year to year and throughout the storage period [43]. For both preharvest and stored maize there were regional differences in fumonisin contamination from a survey during the years 1999 to 2003. Moreover, the fumonisin level varied from one year to the other. The species of *Fusarium* most often isolated from maize were *F. verticillioides* (68%) and *F. proliferatum* (31%). Infection with *F. verticillioides* resulted in fumonisin levels up to 12,000 µg kg^−1^ in 1999–2000, up to 6700 µg kg^−1^ in 2000–2001 and up to 6100 µg kg^−1^ in 2002–2003. The fumonisin contamination of maize was highest in the two southern zones of Benin [44].

Altogether, 26 maize samples from village farms and a large market were analysed just after harvest in Burkina Faso. The average fumonisin content was 130 µg kg^−1^, with a range of 10–450 µg kg^−1^, while the fumonisin content of 26 maize samples stored for one year was 1170 µg kg^−1^, with a range of 110–3120 µg kg^−1^. When 72 samples of maize from the large market in Bobo Dioulasso were analysed, the average fumonisin content was 2900 µg kg^−1^, with a range of 130–16,040 µg kg^−1^ [45]. Analysis of 122 samples, mainly from maize and groundnuts collected in Burkina Faso, included analysis of fumonisin B_1_. The fumonisin incidence was 81% and the median fumonisin contamination was 269 µg kg^−1^ [46]. The European Food Safety Authority (EFSA) has established a tolerable daily intake for fumonisins at 1.0 µg kg^−1^ body weight per day.

In Cameroon, maize samples harvested at three different stages of maturity (80, 85 and 90 days after sowing) followed by drying in the sun or in a barn for one, two or three months, were analysed for fumonisin. All samples were contaminated with fumonisins, in concentrations of 10 to 5990 µg kg^−1^. One week of drying followed by one month of storage increased fumonisin contamination. The drying period in the sun is recommended to be at least two weeks before the storage period. Processing the maize significantly reduced the fumonisin contamination [47]. Altogether, 210 samples of edible non-timber forest products (ENTFP) were collected from farmers and local markets in Cameroon. When the samples were analysed, 53% of them were contaminated with fumonisins, but in only 5% of them was the content above 1000 µg kg^−1^. This is the first report of fumonisin pollution of ENTFP in the Congo Basin forests. The findings will be used in education of farmers and other stakeholders in the country [48]. Samples of maize were collected from 72 farmers’ stores in the humid forests and Western Highlands of Cameroon. Analyses detected fumonisin levels of 300 to 26,000 µg kg^−1^ in the maize. Prolonged storage over 2 to 4 months reduced the fumonisin level in maize [49]. Analysis of mycotoxins in food commodities revealed that maize was contaminated with fumonisins. When 82 samples of dried food commodities were analysed, 41% of the samples contained fumonisins, most at high levels. The average level of FB_1_ in maize was 3684 µg kg^−1^, with a range of 37 to 24 225 µg kg^−1^ [50].

Maize samples from markets and processing plants in Accra, Ghana were analysed. Fourteen samples contained fumonisins in the range of 14 to 4222 µg kg^−1^. A sample from mouldy kernels contained 52,670 µg kg^−1^ fumonisins [51]. The FB_1_ content of domestic poultry feed was in the range of 500 to 4600 µg kg^−1^ (Table 2), which indicates that samples from the feed lots should be analysed before marketing. [52].

A survey during one year in Ogun State, West Nigeria revealed that 13% of the maize samples contained fumonisins above the limit set by the Regulatory Authorities [53]. In North Central Nigeria the fumonisin contamination was 50–8400 µg kg^−1^ in maize harvested from the fields, 50–8150 µg kg^−1^ in maize collected from stores, and 10–6150 µg kg^−1^ in maize offered on the markets [54]. Maize is an important source of nutrition for infants (1–4 years) and young children (5–12 years) in Nigeria. Therefore, fumonisin contamination of baby food is a challenge [55]. Measures to control the fumonisin contamination of domestic food include education of the population on the danger of mycotoxin-contaminated food, early harvesting of maize, rapid drying of the crop, sorting, sanitation, use of improved storage structures, smoking of the maize storage, insect control, the use of botanicals, pesticides, and fumigation [56].

Altogether 55 samples of raw maize and 12 samples of sorghum were collected in the market in Togo and analysed for fumonisins. Of the maize samples, 88% contained fumonisins in concentrations from 101 to 1831 µg kg^−1^, and 67% of the sorghum samples contained from 81.5 to 361 µg kg^−1^ fumonisins [57].

When maize samples offered for sale at Lubumbashi, the second largest city in the Democratic Republic of Congo, were analysed, a fumonisin content of up to 603,000 µg kg^−1^ was detected. The population consuming the contaminated maize are exposed to a health risk [58]. Samples of maize offered for sale at another local market had fumonisin contents ranging from 20 to 9400 µg kg^−1^. The results should be considered, and strategies to reduce mycotoxin threat to consumer health in the country must be developed [59].

### 2.4. Fumonisins in Eastern African Countries

The most common pathogen on maize in Ethiopia is *F. verticillioides*. When 200 samples from the 20 maize growing areas in Ethiopia were analysed, 77% were positive for fumonisins in concentrations ranging from 25 to 4500 µg kg^−1^, and the mean fumonisin content was 348 µg kg^−1^ [60]. The resistance in native Ethiopian cultivars and the biocontrol potential of *Trichoderma* species were evaluated both in vitro and in field trials. A total of seven species of *Fusarium*, with *F. verticillioides* as the most prevalent, were identified from samples collected during 2017–2018. The fumonisin content varied from 105 to 5460 µg kg^−1^. The average fumonisin content was higher in recently harvested maize (2509 µg kg^−1^), than in maize stored for 3 months (1668 µg kg^−1^, Table 3) [61].

In the East Hararghe district of Ethiopia, sorghum is the main staple food. When samples of sorghum grain from the district were analysed for fumonisin content, the fumonisin levels varied from 907–2041 µg kg^−1^; freshly harvested sorghum had the lowest content [62]. Another analysis of barley, sorghum and teff (*Eragrostis tef*) detected fumonisins only in sorghum, and at concentrations up to 2117 µg kg^−1^ [63].

In Kenya, 86.7% of the maize farmers are smallholders, who store their maize crop in polypropylene bags (PP bags). In a survey of maize without symptoms, the FB_1_ levels ranged from 22 to 1348 µg kg^−1^. In Malawa district, the average FB_1_ content in maize after 4-, 8- and 12-weeks storage were 56, 80 and 317 µg kg^−1^, respectively. In the Tongaren district, the FB_1_ content in maize after 4-, 8- and 12-weeks storage were 41, 179 and 590 µg kg^−1^, respectively [64]. After three months in storage, the *Fusarium* contamination was 74.6 % higher in maize stored in PP bags than in maize stored in triple-layer hermetic bags (PICS^TM^) bags. The fumonisin content was 57.1% lower in PICS^TM^ bags than in PP bags [65]. In an assessment of fumonisin contamination of maize in Western Kenya, samples were analysed 2 and 4 months after harvest. Of the maize samples, 87% had detectable fumonisins and 50% of the samples had a fumonisin content above the regulatory level of 1000 µg kg^−1^ [66]. The potential for reducing fumonisin exposure to consumers by sorting maize in the unregulated food systems of Kenya was studied by Ngure. When analysed, almost half (48%) of the 204 samples originating from Western Kenya contained fumonisins above 2000 µg kg^−1^. Analysis of 24 samples from Meru County, Eastern Kenya confirmed that fumonisin contamination was equally common in that part of Kenya. Density sorting to eliminate 31% of the light maize reduced fumonisin contamination in maize lots by 33% [67]. Maize collected from smallholders in the western districts of Kenya had a high incidence of kernel infections of fumonisin producing fungi. When 197 samples were analysed, 47 % of the samples had a FB_1_ content above the detection limit of 100 µg kg^−1^, but only 5% had a FB_1_ contamination above 1000 µg kg^−1^. Four heavily contaminated samples had FB_1_ content from 3600 to 11,600 µg kg^−1^ [68]. A survey of 255 smallholder farmers in Western Kenya showed that fumonisin contamination depended on the cropping systems and pre-harvest agronomic practices (Table 4). Fumonisin levels were positively correlated with the application of diammonium phosphate (DAP) [69].

Maize cultivation is promoted in Rwanda. Analysis of maize samples revealed a fumonisin content ranging 0 to 2300 µg kg^−1^, but only one had a fumonisin content above 2000 µg kg^−1^ [70]. Risk factors due to mycotoxins in food and feed were assessed in another study in Rwanda. Fumonisins were determined in 3328 feed samples, obtained in 2017 from feed processors, feed salesmen, dairy farmers, and poultry farmers. The mean fumonisin levels were 1520 µg kg^−1^ (median 710 µg kg^−1^) for 10 feed processors, 1210 µg kg^−1^ (median 560 µg kg^−1^) for 68 feed salesmen, 1480 µg kg^−1^ (median 760 µg kg^−1^) for 225 dairy farmers and 1030 µg kg^−1^ (median 470 µg kg^−1^) for 309 poultry farmers [71]. Maize samples collected from 15 districts in the country were analysed, and the fumonisin content varied from 0 to 2300 µg kg^−1^. Only one of the samples contained more than 2000 µg kg^−1^ fumonisins [72].

Among East African countries, the highest fumonisin contamination of maize (18,184 µg kg^−1^) has been reported from Tanzania. The East African Community has set the regulatory limit for fumonisins at 2000 µg kg^−1^ [71]. Examination of 114 children under 36 months of age was performed to determine if dietary mycotoxin intake may compromise children growth in the country. The data revealed that their exposure to fumonisins was associated with children being underweight. Fumonisins exposure during 24–36 months of age may have contributed to the high growth impairment rate among children in the Haydom town, Manyana region of Tanzania [73]. Samples of the maize harvest were collected from 120 household in rural areas, and fumonisin levels were determined. In 52% of the samples, fumonisins were detected in concentrations up to 11,048 µg kg^−1^, and 15% of the samples exceeded 1000 µg kg^−1^ [74]. Fumonisin exposure was determined for 215 infants, and of those, 191 consumed maize. Their fumonisins intake varied from 21 to 3201 µg kg^−1^. For 26 infants, the fumonisin exposure exceeded the maximum daily intake of 2 mg/kg bodyweight, and at the age of one year, the babies were 1.3 cm shorter and 328 g lighter than the control group [75]. Analysis of maize samples from 300 household stores detected FB_1_ in 73% of them in concentrations ranging from 16 to 18,184 µg kg^−1^ and FB_2_ in 48 % of the samples in concentrations of 178–38,217 µg kg^−1^ [76]. A survey of 120 farms in three agro-ecological zones of Tanzania revealed that 85% of the maize samples were positive, with fumonisin content in the range of 49–18,273 µg kg^−1^. There were significant differences in fumonisin contamination between maize harvested in the three agro-ecological zones of the country [77]. Maize offered for sale at local markets in Tanzania and in the Democratic Republic of Congo were analysed, and the fumonisin content varied from 20 to 9400 µg kg^−1^ [59].

A survey in Uganda revealed that maize contained from 270 to 10,000 µg kg^−1^ of fumonisin. Maize from high altitudes had the significantly highest fumonisin content, with a mean of 4930 µg kg^−1^, while maize from the mid-altitude moist zone contained on the average 4530 µg kg^−1^, and maize from the mid-altitude dry zone contained on the average 4500 µg kg^−1^. Intercropping, delayed harvesting and drying maize in the fields increased fumonisin content, while crop rotation and seed treatment reduced the contamination [78]. The high frequency of toxigenic *F. verticillioides* in stored maize indicated that most people in Uganda are exposed to high amounts of fumonisins. Among infants, exposure to fumonisins is higher than the tolerable daily intake [79]. The consumers in the northern region of Uganda are exposed to high frequency of toxigenic strains of the pathogen [80].

### 2.5. Fumonisins in Southern African Countries

In Botswana, traditional malt, wort, and beer samples were collected from three villages around Gaborone and analysed for fumonisins and other mycotoxins. Of the 46 malt samples, fumonisin B_1_ was detected in three with concentrations ranging from 47 to 1316 µg kg^−1^ [81]. In a survey of sorghum, peanut butter, and pulses in Botswana, FB_1_ was detected in 36% of the food and feed samples collected. In maize samples, 85% were contaminated with concentrations of FB_1_ ranging from 10 to 432 µg kg^−1^. No fumonisins were detected in peanuts and beans [82].

In a survey of rural households in Malawi, all maize samples analysed contained fumonisin. Maize samples from the Shamva district contained FB_1_ in the range of 10.43 µg kg^−1^ to 432.32 µg kg^−1^, with a median content of 292.15 µg kg^−1^. The average content of FB_1_ in samples from the Makoni district was in the range of 13.84–606.64 µg kg^−1^, with a median of 360.18 µg kg^−1^. In the Shamva district, the probable daily intake of FB_1_ ranged between 0.14 and 5.76 µg kg^−1^ body weight, while in the Makoni district the daily intake of FB_1_ ranged between 0.18 and 8.09 µg kg^−1^ body weight/day [83]. From the three Malawi regions Northern, Central and Southern, maize was sampled for fumonisins analysis. Maize from the Southern region was heavily contaminated with fumonisin, and maize from the Central region also contained high levels, while maize from the Northern Region had the lowest contamination. The maximum concentration of fumonisins in the survey was 7000 µg kg^−1^ [84]. From 31 primary schools, 496 school children under the School Meals Programme were studied to determine the fumonisin exposure. Over 95% of the schools used maize as the main ingredient in the porridge served to the children. The fumonisins intake of the children was estimated to be 6.0–9.2 µg kg^−1^ bodyweight/day, which exceeds the European Union recommended safety level standards for children [85]. In a domestic survey, all beer samples analysed contained fumonisin. The combined contents of FB_1_ and FB_2_ were 1745 ± 1294 µg kg^−1^, and the combined contents of FB_1_, FB_2_ and FB_3_ were 1898 ± 1405 µg kg^−1^ (mean and standard deviation) [86].

A study of fumonisin contamination of maize was reported from Mozambique. Altogether, 122 samples, mainly from maize and groundnuts harvested in the country, were analysed for fumonisins content. Fumonisin contamination was detected in 92% of the maize samples, and the median FB_1_ content was 869 µg kg^−1^ [46].

Traditional Namibian fermented beverage is based on sorghum and pearl millet. A total of 105 samples from the beverage were analysed for fumonisin. The maximum level of FB_1_ was 3060 µg kg^−1^ and for FB_2_ the maximum level was 123 µg kg^−1^ [87].

South Africa is one of the major maize producing countries in Africa. *Fusarium verticillioides* is the most prevalent pathogen in maize, which is a major stable food in rural communities in South and Southern Africa [26]. Natives of the Centane region of the Eastern Cape province consume from 344 to 474 g maize daily. The mean level of fumonisins in home-grown maize was determined to be 1142 µg kg^−1^, while in commercial maize the mean level was 222 µg kg^−1^ [88]. Results from a survey revealed that FB_1_ is the most prevalent mycotoxin in North-Western districts of South Africa (Table 5). The incident rates were 100% on small scale farms and 98.6% on large commercial farms. The results demonstrated that maize, especially from small-scale farmers, contributes to the dietary exposure to mycotoxins [89].

Field samples were collected at homesteads in South Africa and analysed for fungal DNA and fumonisin. From 40 samples collected in Easter Cape in 2005, a high correlation between levels of fumonisins and fungal DNA (R^2^ = 0.8303) was determined. In 126 samples from four provinces collected in 2007, a similar correlation (R^2^ = 0.8658) was found [16]. Fumonisin (FB_1_) contamination was determined in maize and porridge consumed by the rural population of Limpopo Province, South Africa. The fumonisin levels were significantly higher in maize (101–53,863 µg kg^−1^) as compared with the fumonisin levels in porridge (0.2–20 µg kg^−1^) and faecal samples (0.3–464 µg kg^−1^). Further research indicated that a high proportion of the fumonisins is destroyed in the process of making porridge from the maize [90]. Altogether, 114 samples from maize grown by rural subsistence farmers in two districts of northern South Africa were analysed for fumonisins during two seasons. In 2011, the fumonisin content ranged from 12 to 8514 µg kg^−1^, and in the 2012 season it ranged 11 to 18,924 µg kg^−1^ [91]. Statistical data from South Africa indicated that environmental conditions affected the fumonisin contamination of maize. Cultivars DKC80-12B and LS8521B had some resistance to fumonisin contamination. There was no significant correlation between colonization of maize grain and the data for temperature and precipitation, but there was a tendency that increased maximum temperature led to more fumonisin contamination of the maize [92].

Studies in the Transkei region of the Eastern Cape province of South Africa implicated *F. verticillioides* in the development of human oesophageal cancer [93]. Fumonisin contamination of traditional Xhosa maize-beer was determined in the two areas Centane and Bizana of the Transkei region in South Africa. All samples contained fumonisins, and while the content of FB_1_ was 38 to 1066 µg L^−1^, the total fumonisins (FB_1_, FB_2_ and FB_3_) ranged from 43 to 1329 µg L^−1^ [94]. A survey among subsistence farmers in South Africa confirmed that *F. verticillioides* was more common in maize fields than *F. subglutinans* and *F. proliferatum*. Analysis of samples revealed a fumonisin contamination of 0 to 21,800 µg kg^−1^, with large variation among regions. The fumonisin level was highest in northern KwaZulu-Natal, where 52% of the samples in 2006 contained more than 2000 µg kg^−1^, and in 2007 when 17% of the samples contained fumonisins above that level. Also, many samples from Zululand, Limpopo and Eastern Cape province contained fumonisins above the 2000 µg kg^−1^ set by the Food and Drug Administration in USA [95]. A study of 92 commercial compound feeds in South Africa revealed that fumonisins were present in a range of 104–2999 µg kg^−1^. Apart from a few lots, the fumonisin levels were considered safe in livestock production [50]. Fumonisins were detected in cowpea cultivars harvested in South Africa. Analysis revealed FB_1_ at concentrations between 0.12–0.61 µg kg^−1^. This was the first report of fumonisins in cowpea. When cowpea from Benin were analysed, no fumonisins were detected [96]. Morogo is a traditional grain-based food with vegetables. The fumonisin contamination of morogo is a health problem in parts of South Africa [97]. In a community the high rate of oesophageal cancer was associated with consumption of home grown maize due to fumonisin contamination.

Home grown maize and maize porridge in the Centane magistral area were analysed for fumonisin B_1_, B_2_ and B_3_. Porridge consumption of 0.34 kg/body weight day^−1^ resulted in a fumonisin exposure of 6.73 (3.90–11.6) µg kg^−1^ body weight day^−1^. Removal of infected/damaged kernels reduced the fumonisin exposure by 62%. This intervention has the potential to improve food safety and health for the subsistence farming communities, which consume maize contaminated with fumonisins [98]. Exposure to fumonisins for a person with 60 kg body weight were compared for two communities in the former Transkei province. In the Bizana area with relatively low oesophageal cancer incidence, the fumonisin exposure was 3.43 ± 0.15 µg kg^−1^ body weight day^−1^, while in the Centane area with high oesophageal cancer incidence the fumonisin exposure was 8.67 ± 0.18 µg kg^−1^ body weight day^−1^ [99]. Fumonisins were determined in maize from subsistence farmers in two areas. The average incidence of *F. verticillioides* in high quality maize from the Centane area was 16% in both 1997 and 2000 and 32% in 2003, while the incidence in quality maize from the Mbizana area was 11% in both 2000 and 2003. The mean fumonisin content in quality maize from the Centane area was 975 µg kg^−1^ in 2000 and 2150 µg kg^−1^ in 2003. In the Mbizana area, the fumonisin content in fine maize from 2000 was 950 µg kg^−1^, but fumonisin content decreased to 610 µg kg^−1^ in 2003 [12].

In the Lusaka province of Zambia, a survey was conducted to determine the level of fumonisins in maize and maize products. Altogether, 66 samples of maize grain, maize flour and popcorn were collected from farms, markets, street vendors and hammer mills. The highest fumonisin level of 2991 mg kg^−1^ was detected in maize grains obtained from farms, and in maize from a hammer mill with a fumonisin level at 1659 mg kg^−1^ [100]. In field experiments on the medium and the high rainfall zones of Zambia, maize was artificially inoculated with *F. verticillioides*. The occurrence of fumonisins (FB_1_ and FB_2_) in the inoculated maize crop ranged from 0 to 13,050 µg kg^−1^, with an overall mean of 666 µg kg^−1^. Postponing the planting time by 10 or 20 days did not affect the fumonisin contamination, but the delay reduced the maize yield. Maize samples from the high rainfall zone had a low incidence of fumonisins and only 41% of the samples contained fumonisins [101].

In Zimbabwe, 800 maize and 180 small grain (sorghum, pearl millet and finger millet) samples from four agro-ecological zones were analysed for fumonisins at harvest and during storage. Of the maize samples analysed, 54% had a fumonisin content exceeding the EU Regulatory limit of 1000 µg kg^−1^. Less than 10% of the small grain samples had a fumonisin content above the EU Regulatory limit [102]. Analysis of 72 randomly selected maize meal samples revealed that all samples were contaminated with FB_1_ in concentrations between 61.45 and 265.79 µg kg^−1^ (Table 6) while FB_2_ was detected in 56.9% of the samples in concentrations of 13.72 to 76.93 µg kg^−1^ [103].

All the 388 maize samples from rural households in Shamva and Makoni districts of Zimbabwe contained FB_1_. In the Shamva district, the FB_1_ concentrations were from 10.43 to 434.32 µg kg^−1^, with a median content of 292.15 µg kg^−1^, while in the Makoni district the FB_1_ concentrations were from 13.84 to 606.64 µg kg^−1^, with a median concentration of 360.18 µg kg^−1^. In the Shamva district, the probable daily intakes of FB_1_ were in the range of 0.14 to 5.76 µg kg^−1^ body weight, while in the Makoni district the estimated daily intake of FB_1_ were in the range of 0.14 to 8.09 µg kg^−1^ body weight [82]. Subsistence farmers produce maize with rather high fumonisin contamination. From household stores, a total of 95 maize meal samples were randomly collected from subsistence farmers. Agronomic practices and maize intake were investigated, and the fumonisin contents of the maize samples were determined. The FB_1_ intake was calculated to be from 15.0 to 37.2 µg kg^−1^ body weight/day. The exposure to fumonisin, based on maximum tolerable daily intake, was calculated to be 196% for kids under 5 years, 272% for children, 220% for adolescents, 115% for adults and 110% for elderly [104].

### 2.6. Intervention Strategies

Intervention strategies to reduce the impact of fumonisins on African agriculture include development of maize cultivars with resistance to *F. verticillioides*, use of fungicides to control the pathogen and biocontrol by application of microorganisms antagonistic to the pathogen [105]. In the effort to develop maize varieties with resistance to *F. verticillioides* several quantitative trait loci (QTL) and markers for resistance to the pathogen have been developed [106]. Isolates of the biocontrol bacteria *Bacillus subtilis* and *Streptomyces araujoniae* were applied in field experiments where maize plants were sprayed at the end of the vegetative stage (V9) and at the beginning of the reproductive stage (R1). Ten days later all maize ears were inoculated with *F. verticillioides*. Foliar diseases, maize grain yield, *F. verticillioides* incidence and fumonisin contamination of the kernels were recorded [107]. Fungicide treatment reduced foliar diseases, but not infection of the kernels. Application of a fungicide followed by biocontrol bacteria reduced foliar disease, while *F. verticillioides* infection of the kernels was not reduced. Fungicide followed by biocontrol agents reduced the *F. verticillioides* incidence compared to control [108].

## 3. Discussion

Maize has become an important food for humans and feed for domestic animals throughout the African continent. In Sub-Saharan African countries, maize cultivation has grown to a level where maize is the most important cereal crop, and contributes up to 40% of the total daily food intake for humans. Additionally, in North Africa maize cultivation is increasing to contribute to the food supply for the growing human population. Plant breeding to develop varieties adapted to the different climatic zones of Africa has been important in the expansion of maize cultivation on the continent. The fumonisin contamination of maize and other crops has been elucidated during the last three decades. The recent development of sensitive, analytical instruments has made the mycotoxin challenge evident. Fumonisin contamination of maize is increased by stress during the growing season, delayed harvest, and humid storage facilities. Measures to reduce the fumonisin contamination include early harvest, rapid drying of the maize crop, sorting, use of storage facilities and pest control. Quantitative PCR [16,33,109] is a quick and cheaper way to estimate the risk for high fumonisin levels than chromatographic methods [41,89] and it can be used already before harvesting. When maize kernels contain high levels of fumonisin, if is better to use them for animal feed than directly for human food. Education of the farmers is important to reduce the fumonisin contamination of the crops. Potential interventions to reduce the fumonisin contamination include biocontrol and use of clay adsorbents, antioxidants, plant extracts, and essential oils. The increased cultivation of other crops instead of maize and intercropping [69] with maize would also decrease fumonisin intake, while crop rotation would increase yields and diversify food. For subsistence farmers, relevant control measures include sorting of the harvested crop, winnowing and dehulling. With the growing importance of the crop for human consumption in Africa, more research on fumonisin contamination of the maize in the production countries is urgently needed.

## Institutional Review Board

Not applicable.

## Figures and Tables

**Table 1 toxins-14-00419-t001:** Maize production in ten African countries 2008 [30].

Country	Production (Ton)
Nigeria	7,800,000
South Africa	7,338,738
Egypt	7,045,000
Ethiopia	4,000,000
Malawi	3,444,700
Tanzania	3,400,000
Kenya	3,240,000
Mozambique	1,579,400
Zambia	1,366,158
Uganda	1,262,000

**Table 2 toxins-14-00419-t002:** Fumonisin FB_1_ (µg kg^−1^ ) in Poultry Feed Produced in Ghana [52].

Region	Mean FB_1_	Range FB_1_
Acora	2700	800–3100
Ashanthi	1500	300–4600
Western	1200	800–1400
Brong Ahafo	1300	500–1500

**Table 3 toxins-14-00419-t003:** Fumonisins content of recently harvested maize in Eastern Ethiopia during the growing season 2017/18 [61].

Locality	Mean Fumonisin µg kg^−1^	Range Fumonisin µg kg^−1^
Goromuti	2042	316–5102
Gierawa	2790	1475–3586
Tullo	2781	918–5296
Haramaya	2072	294–4850
Meta	2889	827–5394

**Table 4 toxins-14-00419-t004:** Fumonisin levels (µg kg^−1^) in maize from push-pull and non-push-pull cropping systems in five counties in western Kenya [69].

Cropping System	Sampling Size	Proportion of Samples %		Highest Level	
		˂LOD	˂1000	˃1000	
Push-pull					
Kakamega	18	94.4	5.6	0.0	210
Kisumu	21	76.2	19.0	4.8	1439
Migori	34	79.4	14.7	5.9	4471
Siaya	27	88.9	7.4	3.7	1337
Vihiga	16	87.5	12.5	0	145
Non-push-pull					
Kakamega	29	89.7	3.4	6.9	10,412
Kisumu	34	73.5	23.5	2.9	2325
Migori	32	81.3	0.0	18.8	50,769
Siaya	28	71.4	14.3	14.3	9925
Vihiga	16	62.5	25.0	12.5	5177

Abbreviation: LOD, lower limit of detection.

**Table 5 toxins-14-00419-t005:** Summary of mycotoxin contamination in maize produced by subsistence farmers in South Africa [89].

Fumonisin	Positive	Range µg kg^−1^	Mean µg kg^−1^
FB_1_	100%	28.8–1566.7	672.5
FB _2_	39.8%	12.4–239.0	188.4

**Table 6 toxins-14-00419-t006:** Total mean fumonisins (FB_1_ and FB_2_) and projected daily intake of marketed maize meal in Harare, Zimbabwe [103].

Type of Maize Meal	Total Mean Fumonisin Concentration (µg kg^−1^)	Average Projected Daily Intake (µg kg^−1^ bw/Day)
Meal with maize bran added	342.72	4.37
Roller meal	262.68	3.50
Super refined	94.21	1.70
Meal with wheat bran added	61.45	0.82

## Data Availability

Not applicable.

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
