# Peer review of "Fumonisins in African Countries"

_toxins, 2022, doi:10.3390/toxins14060419_

Round 1
Reviewer 1 Report
The manuscript reviews the reported occurrence of fumonisins in Africa. These types of studies are important to justify research in the field, and play an important role in food and feed safety. Previous work is adequately reviewed, and the review covers almost all, if not all, of the current survey work.
The discussion of the relevance of the survey work is lacking and further discussion of the reviewed material can help a broader audience better understand the importance of this paper. The manuscript reviews fumonisin occurrence based on country. Adding information on the ecosystem and environmental factors will be more helpful than just country.
Is the difference in fumonisin occurrence political? If so, that should be mentioned.
Are there any recent reports of mycotoxicosis in the regions reviewed? The authors cite a paper from the early 1990's that suggests cancer of the esophagus is higher in the region with high mycotoxins. This paper is over 40 years old. Are there any recent reports of mycotoxicosis that are related to the work reviewed in this paper? A discussion on this topic can add alot to the manuscript.
If there is a lack of mycotoxicosis in regions with high levels of fumonisins, that should also be discussed.
The discussion section is a summary of the general topic of mycotoxins. Discussing these results in light of research findings by other groups can enhance the paper.
Overall, this is an important review that can benefit from additional discussion of the results to make it clear how this review impacts real world examples of mycotoxicosis in Africa.
Author Response
Reviewer 1
Comments and Suggestions for Authors
The manuscript reviews the reported occurrence of fumonisins in Africa. These types of studies are important to justify research in the field, and play an important role in food and feed safety. Previous work is adequately reviewed, and the review covers almost all, if not all, of the current survey work.
The discussion of the relevance of the survey work is lacking and further discussion of the reviewed material can help a broader audience better understand the importance of this paper. The manuscript reviews fumonisin occurrence based on country. Adding information on the ecosystem and environmental factors will be more helpful than just country.
- we have revised the manuscript and added five more tables to the manuscript.
Is the difference in fumonisin occurrence political? If so, that should be mentioned.
- we do not think that it is political.
Are there any recent reports of mycotoxicosis in the regions reviewed? The authors cite a paper from the early 1990's that suggests cancer of the esophagus is higher in the region with high mycotoxins. This paper is over 40 years old. Are there any recent reports of mycotoxicosis that are related to the work reviewed in this paper? A discussion on this topic can add alot to the manuscript.
- The paper of Rheeder has been replaced by a new paper.
If there is a lack of mycotoxicosis in regions with high levels of fumonisins, that should also be discussed.
- we do not know about it.
The discussion section is a summary of the general topic of mycotoxins. Discussing these results in light of research findings by other groups can enhance the paper.
- we have revised the discussion
Overall, this is an important review that can benefit from additional discussion of the results to make it clear how this review impacts real world examples of mycotoxicosis in Africa.
- thank you
Reviewer 2 Report
I have reviewed the review manuscript titled: Fumonisins in African countries.
This article aims to review the fumonisins content among the North, West, South and Eastern African countries through cereal samples especially from maize of exposure of individuals to these fungal toxins for humans and feeds. The information of this manuscript is useful for feeds and food and relevant processing industries. Although, there mentions little methods to reduce the fumonisin contamination such as early harvest, rapid drying of the corn or other crop, sorting, use of pest and facilities control. The manuscript could be adapted by mycotoxin researching fields especially for Fusarium species in the future. I think the manuscript is acceptable after minor revision. The article is not innovative, however, it contains original and interesting information for the maize and other cereal commodities contaminated with fumonisins in African countries.
Abstract is well written upon and high levels of fumonisin contaminated maize in African countries were revealed.
Introduction is well addressed including major mycotoxins causing contamination of feeds and the disease related to animals. The chemical structure information of the majority of fumonisin B1 and A-, B-, C- and P-series of fumonisins were cited. The regulatory fumonisin content limits of The European Union and Food and Drug Authority and EU were mentioned.
Discussion includes a few methods to reduce the fumonisin contamination by decrease stress during the growing season, delayed harvest, and humid storage facilities. Potential methods to reduce the fumonisin contamination were mentioned for maize production African countries.
I am not a native English speaker. The manuscript seems have no major mistakes are detected and the manuscript can be understood except one specie name in reference section should be italic. The results of the Fumonisin producing fungi and the most common and important sources of fumonisin contaminated cereals are described. Reference format is correct.
I enjoyed reading this manuscript; the needs of special groups of mycotoxin and fumonisins producing fungi in search of the cereal science. This manuscript presents some interesting data.

Author Response
Reviewer 2
Comments and Suggestions for Authors
I have reviewed the review manuscript titled: Fumonisins in African countries.
This article aims to review the fumonisins content among the North, West, South and Eastern African countries through cereal samples especially from maize of exposure of individuals to these fungal toxins for humans and feeds. The information of this manuscript is useful for feeds and food and relevant processing industries. Although, there mentions little methods to reduce the fumonisin contamination such as early harvest, rapid drying of the corn or other crop, sorting, use of pest and facilities control. The manuscript could be adapted by mycotoxin researching fields especially for Fusarium species in the future. I think the manuscript is acceptable after minor revision. The article is not innovative, however, it contains original and interesting information for the maize and other cereal commodities contaminated with fumonisins in African countries.
Abstract is well written upon and high levels of fumonisin contaminated maize in African countries were revealed.
Introduction is well addressed including major mycotoxins causing contamination of feeds and the disease related to animals. The chemical structure information of the majority of fumonisin B1 and A-, B-, C- and P-series of fumonisins were cited. The regulatory fumonisin content limits of The European Union and Food and Drug Authority and EU were mentioned.
Discussion includes a few methods to reduce the fumonisin contamination by decrease stress during the growing season, delayed harvest, and humid storage facilities. Potential methods to reduce the fumonisin contamination were mentioned for maize production African countries.
I am not a native English speaker. The manuscript seems have no major mistakes are detected and the manuscript can be understood except one specie name in reference section should be italic. The results of the Fumonisin producing fungi and the most common and important sources of fumonisin contaminated cereals are described. Reference format is correct.
I enjoyed reading this manuscript; the needs of special groups of mycotoxin and fumonisins producing fungi in search of the cereal science. This manuscript presents some interesting data.
- thank you, minor revision has been done.
Reviewer 3 Report
The manuscript has some mistakes. It is important to have a detailed information regarding the mycotoxins on maize, although these information should be provide in tables. And more research should be included on the manuscript, regarding e.g. strategies to mitigate the mycotoxin contaminations in that the regions, the fungus associated with mycotoxins...
The manuscript must be improved.
Author Response
Reviewer 3
Comments and Suggestions for Authors
The manuscript has some mistakes. It is important to have a detailed information regarding the mycotoxins on maize, although these information should be provide in tables. And more research should be included on the manuscript, regarding e.g. strategies to mitigate the mycotoxin contaminations in that the regions, the fungus associated with mycotoxins...
The manuscript must be improved.
- Thank you, The manuscript has been revised based on the comments of reviewer 3 and one new reference and five tables have been added.
Reviewer 4 Report
This work proposes an extensive review on the fumonisin contamination of maize and other crops in the African countries. As such, the matter is of interest, however the paper suffers for two serious limits:
1) In introduction section, information on aim and objective of study is not well definedï¼›
2) some Figures must be selected from previous literature to discuss also the Pollution factors and preventive measures of maize in the African countries (there are several examples published), which has been largely overlooked throughout the paper.
Once the above concerns are fully addressed, the manuscript could be accepted for publication in this journal.
Author Response
Reviewer 4.
Comments and Suggestions for Authors
This work proposes an extensive review on the fumonisin contamination of maize and other crops in the African countries. As such, the matter is of interest, however the paper suffers for two serious limits:
1) In introduction section, information on aim and objective of study is not well definedï¼›
- The aim is included in the introduction
2) some Figures must be selected from previous literature to discuss also the Pollution factors and preventive measures of maize in the African countries (there are several examples published), which has been largely overlooked throughout the paper.
- the manuscript has been revised and five tables have been added in the manuscript.
Once the above concerns are fully addressed, the manuscript could be accepted for publication in this journal.
- thank you.
Round 2
Reviewer 3 Report
The manuscript must be improved.
Author Response
We have improved the manuscript with intervention strategies and added new references.